# Immunogenicity and efficacy of VLA2001 vaccine against SARS-CoV-2 infection in male cynomolgus macaques
Mathilde Galhaut [1], Urban Lundberg [2], Romain Marlin [1], Robert Schlegl[2], Stefan Seidel[2], Ursula Bartuschka[2], Jürgen Heindl-Wruss[2], Francis Relouzat [1], Sébastien Langlois[1], Nathalie Dereuddre-Bosquet [1], Julie Morin[1], Maxence Galpin-Lebreau[1], Anne-Sophie Gallouët[1], Wesley Gros[1], Thibaut Naninck [1], Quentin Pascal[1], Catherine Chapon[1], Karine Mouchain[3], Guillaume Fichet[3], Julien Lemaitre[1], Mariangela Cavarelli [1], Vanessa Contreras[1], Nicolas Legrand [3], Andreas Meinke[2,4] ✉ & Roger Le Grand [1,4] ✉

## Abstract

**Background** The fight against COVID-19 requires mass vaccination strategies, and vaccines inducing durable cross-protective responses are still needed. Inactivated vaccines have proven lasting efficacy against many pathogens and good safety records. They contain multiple protein antigens that may improve response breadth and can be easily adapted every year to maintain preparedness for future seasonally emerging variants. **Methods** The vaccine dose was determined using ELISA and pseudoviral particle-based neutralization assay in the mice. The immunogenicity was assessed in the non-human primates with multiplex ELISA, neutralization assays, ELISpot and intracellular staining. The efficacy was demonstrated by viral quantification in fluids using RT-qPCR and respiratory tissue lesions evaluation. **Results** Here we report the immunogenicity and efficacy of VLA2001 in animal models. VLA2001 formulated with alum and the TLR9 agonist CpG 1018™ adjuvant generate a Th1-biased immune response and serum neutralizing antibodies in female BALB/c mice. In male cynomolgus macaques, two injections of VLA2001 are sufficient to induce specific and polyfunctional CD4+ T cell responses, predominantly Th1-biased, and high levels of antibodies neutralizing SARS-CoV-2 infection in cell culture. These antibodies also inhibit the binding of the Spike protein to human ACE2 receptor of several variants of concern most resistant to neutralization. After exposure to a high dose of homologous SARS-CoV-2, vaccinated groups exhibit significant levels of protection from viral replication in the upper and lower respiratory tracts and from lung tissue inflammation. **Conclusions** We demonstrate that the VLA2001 adjuvanted vaccine is immunogenic both in mouse and NHP models and prevent cynomolgus macaques from the viruses responsible for COVID-19.

## Plain Language Summary

Mass vaccination in response to the COVID-19 pandemic has substantially reduced the number of severe cases and hospitalizations. As the virus continues to evolve and give rise to new variants that cause local outbreaks, there is a need to develop new vaccine candidates capable of stopping the viral transmission. In this study, we explore the immune responses induced by the vaccine candidate VLA2001 in animal models. We highlight the vaccine's ability to induce an immune response capable of blocking the virus and eliminating infected cells. We show that it can protect the host from developing severe disease.

Durable control of the coronavirus disease 2019 (COVID-19) pandemic, caused by the severe acute respiratory syndrome coronavirus 2 (SARS-CoV-2), requires mass vaccination strategies for which the first vaccines became available at the end of 2020. Vaccines approved and in use to date have demonstrated high protective efficacy against infection and clinically manifest disease[1–8]. However, additional vaccines are needed to achieve sufficient global supply. In addition, several of the vaccines in use have limitations. First, vaccines based on adenovirus vectors have been linked

---

in rare cases to a risk of thrombotic thrombocytopenia and mRNA vaccines with a risk of myocarditis and pericarditis[9,10]. Second, several of the available vaccines utilize one or two SARS-CoV-2 Spike (S) proteins, to elicit protective immunity, as displayed in the bivalent mRNA constructions. Although the efficacy of these vaccines was high against the ancestral virus and remained high against several variants of concern (VOC)[11], it dropped precipitously with the emergence of the omicron VOC[12–17], which has a S protein sequence that is much more divergent from the ancestral SARS-CoV-2 than previous VOCs, despite the high efficacy of these vaccines against severe disease and hospitalization. It has therefore been speculated that inclusion of additional antigens in vaccines for induction of broad cellular immunity may offer better protection against clinically significant infection with other variants such as omicron[13,18].

The introduction of an inactivated vaccine may overcome some of the reasons for the vaccine hesitancy observed against vaccines based on current innovative technologies[19]. Several inactivated vaccines are currently in use in Asia, Africa and South America with variable reported efficacy against COVID-19, up to 50%[6,8,20–26]. Two of these inactivated vaccines are adjuvanted by adsorption to aluminum hydroxide (alum), whereas the third contains alum and the TLR 7/8 agonist imidazoquinoline. VLA2001 is formulated with alum and the TLR9 agonist CpG 1018™ adjuvant and is the first inactivated COVID-19 vaccine that has been authorized by a regulatory agency in Europe.

Here we report on the preclinical evaluation of VLA2001, a vaccine intended for active immunization to prevent carriage and symptomatic infection with SARS-CoV-2. We demonstrate the capacity of the vaccine to induce neutralizing antibodies against several VOCs in mice and non-human primates (NHP) and protection in a NHP challenge model.

## Methods
### Ethical and biosafety statement
All mice experiments were conducted in accordance with Austrian law (BGBl No. 114/2012) and approved by the appropriate local authorities. Experimental procedures were reviewed and approved by Valneva's animal welfare committee and performed by trained personnel. The animals were housed at the Valneva Austria GmbH animal facility in groups of five female mice per cage and checked daily. Mice were fed with a commercial mouse chow (Ssniff RIM-H autoclaved, Ssniff Spezialdiäten GmbH, Germany) *ad libitum* and had free access to tap water. Food and water were autoclaved before they were given to the mice. As environmental enrichment mice were provided wood wool as nesting material.

Cynomolgus macaques (*Macaca fascicularis*), aged 43.8-46.3 months at initiation of the immunizations (24 males), naive of experimental procedure and originating from Mauritian AAALAC certified breeding centers were used in this study. All animals were housed in IDMIT facilities (CEA, Fontenay-aux-Roses), under BSL-2 and BSL-3 containment (Animal facility authorization #D92-032-02, Préfecture des Hauts-de-Seine, France) and in compliance with European Directive 2010/63/EU, the French regulations and the Standards for Human Care and Use of Laboratory Animals of the Office for Laboratory Animal Welfare (OLAW, assurance number #A5826-01, US). The protocols were approved by the institutional ethical committee "Comité d'Ethique en Expérimentation Animale du Commissariat à l'Energie Atomique et aux Energies Alternatives" (CEtEA #44) under statement number A20-037. The study was authorized by the Research, Innovation and Education Ministry under registration number APAFIS#24434-2020030216532863v3. Before entry into the study, animals were tested negative for Salmonella, Yersinia, Shigella, Herpes B, Hepatitis B, SIV, STLV, Tuberculosis, Measles, Flu and Rabies. A total number of 30 animals were serologically screened for the presence of anti-S, anti-RBD and anti-N antibodies. For inclusion into the study, selected animals should be cynomolgus macaque species, naive of experimental procedure and seronegative for anti-S, anti-RBD and anti-N antibodies.

### VLA2001 vaccine
VLA2001 is a highly purified inactivated virus derived from a single plaque of the clinical isolate, hCoV-19/Italy/INMI1-isl/2020 (National Institute for Infectious Diseases [INMI], Rome, Italy, GISAID Accession EPI_ISL_410545, Pango lineage B) isolated in January 2020 from a patient who was traveling from China. The virus had been passaged twice on Vero E6 cells at INMI and plaque purified twice to minimize the risk of any adventitious agents at Valneva. The GMP manufacturing vaccine was produced on Vero cells, concentrated using ultrafiltration, inactivated with β−propiolactone (500 ppm), and purified over a sucrose gradient. The virus was further diluted with PBS dependent on the antigen content, as determined by ELISA. Then, VLA2001 was formulated with alum (Alhydrogel, Croda, Denmark) (17 μg $Al^{3+}$/0.1 mL for mice or 0.5 mg $Al^{3+}$/0.5 mL for NHPs) and CpG 1018™ adjuvant, which was added cage-side prior to injection (10 μg CpG 1018/0.1 mL for mice or 1 mg CpG 1018/0.5 mL for NHPs).

### Mouse study design
Six to eight-weeks old female BALB/c mice (Janvier) were used. Female animals were used due to their availability at the time of experiment. Groups of five mice were immunized subcutaneously (s.c.) twice with 100 μL VLA2001 with three-weeks interval. Placebo was both adjuvants at the same concentrations in PBS. Two weeks after the third immunization, blood was collected in BD Microtainer™ tubes, and serum was prepared by centrifugation at room temperature (RT) for 3 min at $12,000 \times g$. Serum samples were heat-inactivated by incubation at 56 °C for 35 min.

### Immunogenicity and Th1/Th2 responses in mice
ELISA plates (Maxisorp, Nunc) were coated with 100 ng protein in PBS and incubated at 4 °C overnight. The plates were blocked with 5% BSA, 0.05% Tween-20 in PBS for 1–2 h at RT. Plates were washed with PBS/0.1 T (PBS with 0.1% Tween-20). Pooled mouse sera (five per group) and the respective mAbs clone 43 (Invivogen, US) and CR3022 (Sino Biological, Germany) for subclass ELISA were diluted in blocking buffer (four-fold dilution), added to 96-well plates, and tested in duplicates by incubation for 1 hour at RT. Plates were then washed with PBS/0.1 T. The secondary antibodies, Goat Anti-mouse IgG-HRP (1:4,000), Goat Anti-mouse IgG1-HRP (1:3,000) and Goat Anti-mouse IgG2a-HRP (1:3,000) were diluted in blocking buffer, added to the 96-well plates, and incubated for 1 h at RT. Plates were washed with PBS/0.1 T and ABTS (Sigma-Aldrich) was added as substrate. After incubation for 30 min, the reaction was stopped by the addition of 1% SDS and the absorbance was read at 405 nm. The half-max titer (the reciprocal of the dilution that corresponds to the mean absorbance between highest and lowest dilution) was determined. For determination of IgG subclasses IgG1 and IgG2a, the subclass responses were calculated to the respective standard curves (clone 43 and CR3022) and presented as ng/mL equivalent of the respective mAb.

### Functional immune responses in mice
The neutralizing antibody titers were assessed with a pseudoviral particle-based neutralization assay. HEK293-hACE2 cells were seeded into 96-well plates ($2 \times 10^4$ cells/well) and incubated at 37 °C (5% $CO_2$) overnight. Heat-inactivated serum pools were twofold diluted in DMEM with 10% FBS starting at 1:10 and mixed with an equal volume (45 μL) of pseudovirus (SCV2-PsV-614G, eEnzyme) with approximately $0.2 \times 10^6$ RLU/mL and incubated for 1 hour at RT. Of the serum pool/pseudovirus mixture, 80 μL were added to the HEK293-hACE2 cells and incubated for 2 h at 37 °C (5% $CO_2$). After washing once with medium and adding fresh medium, cells were incubated for 48 h at 37 °C (5% $CO_2$). Plate and Bright-Glo Luciferase reagent were equilibrated to RT, then 100 μL Bright-Glo Luciferase reagent were added to each well and luminescence was read after 5 min in a Synergy 2 (BioTek). Percent neutralization was calculated compared to the reference (non-immune serum) at the respective dilution.

## Non-human primate study design

Twenty-four male cynomolgus macaques were randomly assigned, with RAND function of Excel software, into three experimental groups (Supplementary Table 1). Male animals were used due to their availability at the time of experiment. The vaccinated High dose group (Group High) received 35 AU of SARS-CoV-2 inactivated VLA2001 vaccine, adjuvanted with 0.5 mg alum and 1 mg CpG 1018™ adjuvant, by i.m. route. The vaccinated Medium dose group (Group Medium) received 7 AU of SARS-CoV-2 inactivated VLA2001 vaccine, adjuvanted with 0.5 g alum and 1 mg CpG 1018™ adjuvant, by i.m. route. Remaining animals were used as controls and received the equivalent volume of PBS by i.m. route. All animals received two injections of the vaccine at weeks 0 and 3, into the right thigh (0.5 mL per dose). Seven weeks following first vaccine injection (day 0), animals were inoculated with $1 \times 10^5$ PFU SARS-CoV-2 WA1 strain (BetaCoV/France/IDF/0372/2020 strain, EPI_ISL_410720 (GISAID ID)) by intranasal (i.n., 0.25 mL in each nostril) and intratracheal (i.t., 4.5 mL into trachea) routes simultaneously[27–31].

For all handling procedures, animals were anesthetized using ketamine hydrochloride (Imalgen® 1000 5 mg/kg) associated with medetomidine hydrochloride (Domitor® 0.05 mg/kg) by i.m. route. Just after the end of the manipulations, atipamezole hydrochloride (Antisedan® 0.05 mg/kg) was administered to induce recovery from anesthesia. Animals were euthanized at day 7 ($n = 4$ per group) or at day 15 ($n = 4$ per group) post SARS-CoV-2 exposure by an intravenous overdose of pentobarbital (Dolethal® 180 mg/kg) into the saphenous vein. Animals were randomly assigned to early (day 7) or late (day 15) euthanasia group before immunizations.

## Evaluation of anti-S, anti-RBD and anti-N IgG antibody responses in cynomolgus macaque sera

The analysis of the antibody responses (IgG) against SARS-CoV-2 was quantified for NHP antibodies targeting SARS-CoV-2 S, N and RBD antigens, using a multiplex approach with the MSD kit "V-PLEX COVID-19 Coronavirus Panel 2 (IgG)" (ref. K15369U), following the manufacturer's instructions. All samples were inactivated before analysis (30 min at 56 °C). Negative and positive control reagents were provided with the kit. Optimal sample dilution was determined by evaluating positive samples (from infected cynomolgus macaques) at several dilutions (1:100 to 1:1600). A 1:500 dilution, providing no saturation of the signal, was then chosen based on the obtained results. Negative samples (obtained from naive cynomolgus macaques) were tested at a 1:100 dilution to evaluate the non-specific signal, as well as inter-individual variability. Quality control was performed by spiking a monoclonal human anti-SARS-CoV-2 S Glycoprotein S1 antibody (clone CR3022; Abcam ref. ab273073) into negative serum samples to compare different inactivation protocols (0.1% Triton; 30 min at 56 °C). Data acquisition was performed on a MESO Quickplex SQ120 imager and the results were analyzed with the associated software (Discovery Workbench).

## Evaluation of anti-SARS-CoV-2 neutralizing IgG antibodies in cynomolgus macaque sera

The analysis of the SARS-CoV-2 neutralizing antibody responses was performed in a VeroE6-TMPRSS2 cell (JCRB Cell Bank)[32] culture assay in presence of live virus. Pre-incubated mixtures (at different dilution ratios) of NHP heat-inactivated sera (56 °C for 30 min) and SARS-CoV-2 virus (strain SK-BMC5/2020 supplied through the European Virus Archive - Global [EVAg] platform) were placed on plated cells. A reference standard control was included (WHO International Standard First WHO International Standard for anti-SARS-CoV-2 human immunoglobulin NIBSC code 20/136, corresponding to pooled plasma obtained from eleven convalescent individuals recovered from SARS-CoV-2 infection). Data acquisition was performed on an Operetta reader (Perkin Elmer), using the colorimetric CellTiter 96® AQueous Non-Radioactive Cell Proliferation Assay following the manufacturer's protocol (Promega, ref. #G5430).

Using XLfit®, the corresponding Neutralizing titers ($NT_{50}$) were calculated using the nonlinear regression model, expressed as the inverse of the serum dilution that inhibited 50% of infection (lower limit of quantification: 5; NIBSC 20/136 $NT_{50}$: 21).

## Evaluation of anti-S VOC ACE-2 competition assay in cynomolgus macaque sera

Antibodies blocking the binding of SARS-CoV-2 S protein to ACE2 were detected with the MesoScale Discovery V-PLEX® SARS-CoV-2 Panel 1 (ACE2) kit (K0082163, K15375U, MSD®) according to the manufacturer's instructions. The antigens included were WA1, B.1.1.7, B.1.351, B.617.2 + AY.4 alt seq 2, B.1.1.529 + BA1, BA.2, BA.2.12.1, BA.2.75 and BA.5 S.

The plates were blocked with 50 μL of blocker A (1% BSA in MilliQ water) solution for at least 30 min at RT shaking at 700 rpm with a digital microplate shaker. During blocking, heat-inactivated serum samples were diluted 1:10 and 1:100. Each plate contained duplicates of a 7-point calibration curve with serial dilution of a reference standard, and a blank well. The plates were then washed three times with 150 μL of the MSD kit Wash Buffer, blotted dry, and 25 μL of the diluted samples were added to the plates and set to shake at 700 rpm at RT for at least 2 hours. The plates were again washed three times and 25 μL of SULFO-Tagged human ACE2 protein was added to each well and incubated shaking at 700 rpm at RT for at least 1 hour. Plates were then washed three times and 150 μL of MSD GOLD Read Buffer B was added to each well. The plates were read immediately after on a MESO QuickPlex SQ 120 machine. Electro-chemiluminescence (ECL) signal was recorded and results expressed as % of inhibition ($\% \ of \ inhibition = 1 - \left( \frac{average \ sample \ ECL \ signal}{average \ ECL \ signal \ of \ calibrator \ 8(diluent \ only)} \right) * 100$).

## Antigen specific T cell assays in cynomolgus macaque PBMCs

IFN-γ ELISpot was performed using the Monkey IFN-γ ELISpot PRO kit (Mabtech, ELISpot Pro: Monkey IFN-γ (ALP), #3421M-2APT) according to the manufacturer's instructions. PBMCs were stimulated for 18 hours with S1, S2, RBD or N overlapping pools of 15-mer peptides overlapping by 11 amino acids at a final concentration of 2 μg/mL. Peptide pools S1 and S2 covering the SARS-CoV-2 S protein sequence ($n = 158$ and 157 respectively, 1273 amino acids in total), the SARS-CoV-2 RBD sequence ($n = 53$, 223 amino acids in total) and the SARS-CoV-2 N sequence ($n = 102$, 419 amino acids in total) were synthesized by JPT Peptide Technologies (Berlin, Germany).

T cell responses were also characterized by measuring the frequency of PBMCs expressing IL-2 (PerCP-Cy5.5, MQ1-17H12, Becton Dickinson), IL-17a (AF700, N49-653, Becton Dickinson), IFN-γ (V450, B27, Becton Dickinson), TNF-α (BV605, Mab11, BioLegend), IL-13 (BV711, JES10-5A2, Becton Dickinson), CD69 (PE, FN50, BioLegend), CD137 (APC, 4B4-1, Becton Dickinson) and CD154 (FITC, TRAP1, Becton Dickinson) upon stimulation with the peptide pools. The CD3 (APC-Cy7, SP34-2, Becton Dickinson), CD4 (BV510, L200, Becton Dickinson) and CD8 (PE-Vio770, BW135/80, Miltenyi Biotec) antibodies were used as lineage markers. One million PBMCs were cultured in complete medium (RPMI1640 Glutamax + , Gibco; supplemented with 10% FBS), supplemented with co-stimulatory antibodies (FastImmune CD28/CD49d, Becton Dickinson), and stimulated with S, RBD or N sequence overlapping peptide pools as above at a final concentration of 2 μg/mL. Brefeldin A was added to each well at a final concentration of 10 μg/mL and the plate was incubated at 37 °C, 5% $CO_2$ for 18 hours. Next, cells were washed, stained with a viability dye (LIVE/DEAD fixable Blue dead cell stain kit, ThermoFisher), and then fixed and permeabilized (Cytofix/Cytoperm, BD). Fixed and permeabilized cells were stored at −80 °C before the staining procedure. After thawing of fixed and permeabilized cells, antibody staining was performed. After 30 min of incubation at 4 °C in the dark, cells were washed in BD Perm/Wash buffer and then acquired on the LSRII flow cytometer (BD). Analysis was performed on FlowJo v.10 software[29].

## Virus quantification in cynomolgus macaque fluids

Upper respiratory tract (nasopharyngeal and tracheal) specimens were collected with swabs and placed in viral transport medium (CDC, DSR-052-01). Tracheal specimens were collected by insertion of the swab above the tip of the epiglottis into the upper trachea at approximately 1.5 cm of the epiglottis. Lower respiratory specimens were collected from BAL. Briefly, an endotracheal tube was inserted using a laryngoscope to visualize the epi-glottis and larynx after application of 2% lidocaine. A volume of 50 mL of 0.9% NaCl was injected before compressing the diaphragm to collect the lavage. Viral genomic RNA quantification was performed by RT-qPCR, using RdRp-IP4 primers and probe[33] and a standard curve calibrated using SARS-CoV-2 standard (COV019, Exact Diagnostic). The lower limit of detection was 2.67 $\log_{10}$ copies/mL and the lower limit of quantification was 3.67 $\log_{10}$ copies/mL. SARS-CoV-2 E gene subgenomic mRNA (sgRNA) were quantified by RT-qPCR using primers and probe[34,35]: leader-specific primer sgLeadSARSCoV2-F CGATCTCTTGTAGATCTGTTCTC, E-Sarbeco-R primer ATATTGCAGCAGTACGCACACA and E-Sarbeco probe HEX-ACACTAGCCATCCTTACTGCGCTTCG-BHQ1. The lower limit of detection was 2.87 $\log_{10}$ copies/mL and the lower limit of quantification was 3.87 $\log_{10}$ copies/mL.

## Evaluation of pulmonary lesions by chest-CT imaging in cynomolgus macaques

All image acquisitions were performed on the Digital Photon Counting (DPC) PET/CT system (Vereos-Ingenuity, Philips) implemented in BSL-3 containment. During imaging sessions, animals were maintained sedated under isofluorane 1–2% in supine position on a patient warming blanket (Bear Hugger, 3 M) with cardiac rate, oxygen saturation and temperature monitoring.

The X-ray computed tomography (CT) was performed under breath-hold. The CT detector collimation used was $64 \times 0.6$ mm, the tube voltage was 120 kV and intensity of about 150 mAs. Automatic dose optimization tools (Dose Right, Z-DOM, 3D-DOM by Philips Healthcare) regulated the intensity. CT images were reconstructed with a slice thickness of 1.25 mm and an interval of 0.25 mm.

Images were analyzed using INTELLISPACE PORTAL 8 (Philips healthcare). Pulmonary lesions were defined as Ground Glass Opacity, Crazy-paving pattern or consolidation as previously described[36].

## Pulmonary histopathology in cynomolgus macaques

At necropsy, cranial and caudal lobes of macaque lungs were fixed by immersion in 10% formalin solution for 24 hours. Formalin fixed samples were paraffin embedded (FFPE) with vacuum inclusion processor (Excelsior, Thermo), cut in 5 µm (Microtome RM2255, Leica) slices, mounted on coated glass slides (Superfrost + , Thermo) and stained with haematoxylin and eosin (H&E) with automated staining processor (Autostainer ST5020, Leica).

Each slide was scored blindly in 20 different spots at ×20 magnification by a certified veterinary pathologist (DVM, DESV-AP). On each spot, 5 different parameters were assessed: Septal cellularity, Septal fibrosis, Type II pneumocytes hyperplasia and alveolar macrophages and neutrophils. A systematic histopathology scoring was used and described in Supplementary Table 2. Each score was then cumulated for each assessed field of view for cranial and caudal lobes.

## Statistics and reproducibility

Data were collected using classical Excel files. Data analysis was performed on Prism v9.2.0 (Graphpad) using non-parametric Kruskal-Wallis and Dunn's multiple comparisons tests. Duplicates were performed for RT-qPCR, ELISA and ELISpot. For mice experiment, 5 animals per group were included. For NHP experiment, 8 animals per group were used.

## Reporting summary

Further information on research design is available in the Nature Portfolio Reporting Summary linked to this article.

## Results

### The VLA2001 vaccine induced high levels of anti-SARS-CoV-2 antibodies in mice

VLA2001 is a highly purified inactivated virus derived from a single plaque of the clinical isolate hCoV-19/Italy/INMI1-isl/2020 (National Institute for Infectious Diseases, Rome, Italy, GISAID Accession EPI_ISL_410545; Pango lineage B). VLA2001 was formulated with alum (17 µg $Al^{3+}$/0.1 mL) and CpG 1018™ adjuvant, which is used in the US FDA- and EMA-approved vaccine HEPLISAV-B®[37] and was added cage-side prior to injection (10 µg CpG 1018/0.1 mL).

The immunogenicity of VLA2001 was tested in female BALB/c mice in two independent experiments with dosages ranging from 0.3 to 35 antigen units (AU), corresponding to 0.02 to 2.1 µg total viral protein. VLA2001 formulated with alum, with and without CpG 1018™ adjuvant was given subcutaneously two times with three weeks interval and sera taken two weeks after the second immunization were analyzed. The immune response was analyzed with ELISA for antibodies that bind the homologous S1 subunit and receptor binding domain (RBD) of S glycoprotein and N protein. Increases in antibody titers against the S1 and RBD parts of the S glycoprotein were obtained for VLA2001 formulated with alum and CpG 1018™ adjuvant as compared to vaccine formulated with alum alone (Fig. 1a, b, d, e). The presence of CpG 1018™ adjuvant increased also Nucleoprotein (N)-binding antibodies, with the largest effect observed at the lowest immunization dose (0.3 AU; Fig. 1c, f). Thus, for all dose levels of VLA2001, an increased immune response as determined by ELISA was seen for VLA2001 when formulated with CpG 1018™ adjuvant and alum as compared to alum alone, indicating a superior immune response due to the presence of CpG 1018™ adjuvant. In addition, the VLA2001 vaccine containing CpG 1018™ adjuvant induced a higher proportion of IgG2a antibodies, indicative of a Th1-biased immune response in mice, whereas the VLA2001 vaccine formulated with alum only induced a higher proportion of IgG1 antibodies, consistent with a Th2-biased immune response (Fig. 1g). Functional antibodies in mouse immune sera after immunization with VLA2001 were assessed in a pseudoviral particle-based neutralization assay. The serum pool from mice immunized with 35 AU of VLA2001 formulated with alum and CpG 1018™ adjuvant generated the highest 50% neutralizing titers ($NT_{50}$) of 246, whereas VLA2001 formulated with alum alone at the same antigen dose (35 AU) generated a $NT_{50}$ of 52 (Fig. 1h). Thus, at 35 AU, VLA2001 formulated with alum and CpG 1018™ adjuvant generated an immune response with an approximately 5-fold higher neutralizing titer which is a fold increase in the same range as seen for the immune response measured by ELISA against the S glycoprotein.

### The VLA2001 vaccine induced robust B and T cell responses in cynomolgus macaques

NHPs are particularly valuable to assess human vaccine candidates, because the similarities of the immune systems between the two species make immunogenicity and efficacy studies in NHP highly predictive of vaccine effects in humans[38–40]. We have tested the vaccine VLA2001, formulated with alum and CpG 1018™ adjuvant (0.5 mg $Al^{3+}$/0.5 mL and 1 mg CpG 1018/0.5 mL), in two groups of eight male cynomolgus macaques injected either with a high dose (35 AU corresponding to 2.1 µg of total virus protein, Group High) or medium dose (7 AU corresponding to 0.3 µg of total virus protein, Group Medium) by intramuscular (i.m.) route on days 0 and 21 (Fig. 2a). A third group of eight animals received PBS at the same time points as control. All VLA2001 vaccinated animals raised serum IgGs against the homologous S protein as early as week 2 post first vaccine injection ($p = 0.0078$ for each vaccinated group compared to controls, non-parametric Kruskal-Wallis and Dunn's multiple comparisons tests; n = 8 per group) with slightly increased levels in Group High compared to Group Medium which did not reach statistical significance (Fig. 2b and Supplementary Fig. 1). Following booster injection, anti-S IgG significantly increased in both groups to similar levels ($p = 0.0078$ for each vaccinated group compared to controls, non-parametric Kruskal-Wallis and Dunn's multiple comparisons tests; $n = 8$ per group). Profiles of anti-RBD IgG

**Fig. 1 | Immunogenicity of VLA2001 in mice.**
Doses used for immunizations were from 0.3 to 3.0
arbitrary units (AU) (**a–c, g**) or 3.0 to 35 AU (**d–f, h**).
Antibody titers were determined by ELISA using
pooled sera from each group. Plates were coated
with either S1 (**a, d**), RBD (**b, e**) or N-protein (**c, f**).
Gray bars depict serum pools from mice immunized
with VLA2001 formulated with alum and black bars
show VLA2001 formulated with alum and CpG
1018™ adjuvant. Endpoint titers were determined
using 3× the absorbance (405 nm) of the blanks as
cut-off. The dotted lines refer to the limit of detec-
tion (LoD) 50, values below LoD were imputed with
25. **g** The IgG subclass responses (filled bars: IgG1,
hatched bars: IgG2a) were determined using S pro-
tein specific mAbs with different IgG subclasses as
references. **h** Pseudovirus neutralization assay was
performed using MLV particles with SARS-CoV-2 S
protein (614 G) and firefly luciferase as reporter. The
dotted line depicts the LoD 20, values below LoD
were imputed with 10.

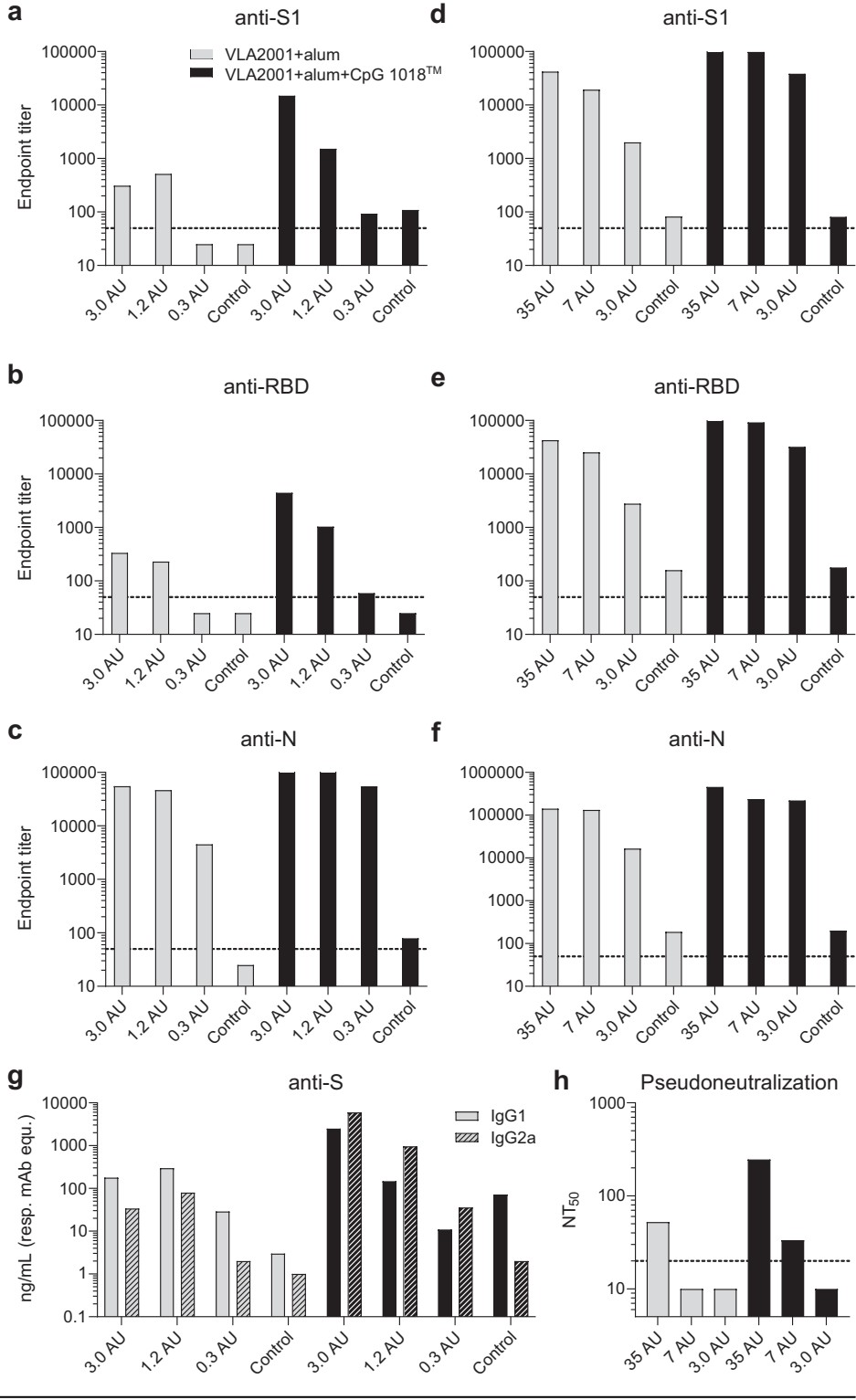

responses paralleled the anti-S responses. One week after the boost, both
groups had developed similar levels of specific IgGs. VLA2001 also induced
high levels of N-binding IgG that were similar in both groups of macaques
(Fig. 2b and Supplementary Fig. 1). This first set of analyses indicated that
two injections of the 7 AU dose were sufficient to induce high levels of
SARS-CoV-2 specific antibodies exceeding the levels reported using iden-
tical immune-assays in convalescent patients[41,42] and in convalescent
macaques from our previously reported studies[28,29,43]. The VLA2001 vaccine

induced antibodies were also able to neutralize SARS-CoV-2 infection in
cultured Vero cells. Two weeks following the boost, serum neutralization
activity was significantly higher in both vaccinated groups compared to
controls ($p < 0.0001$ and $p = 0.0230$ for Groups High and Medium respec-
tively, non-parametric Kruskal-Wallis and Dunn's multiple comparisons
tests; $n = 8$ per group) and also higher than the first WHO international
standard for anti-SARS-CoV-2 immunoglobulins (Fig. 2c and Supple-
mentary Fig. 2). These antibodies have the potential to cross-neutralize

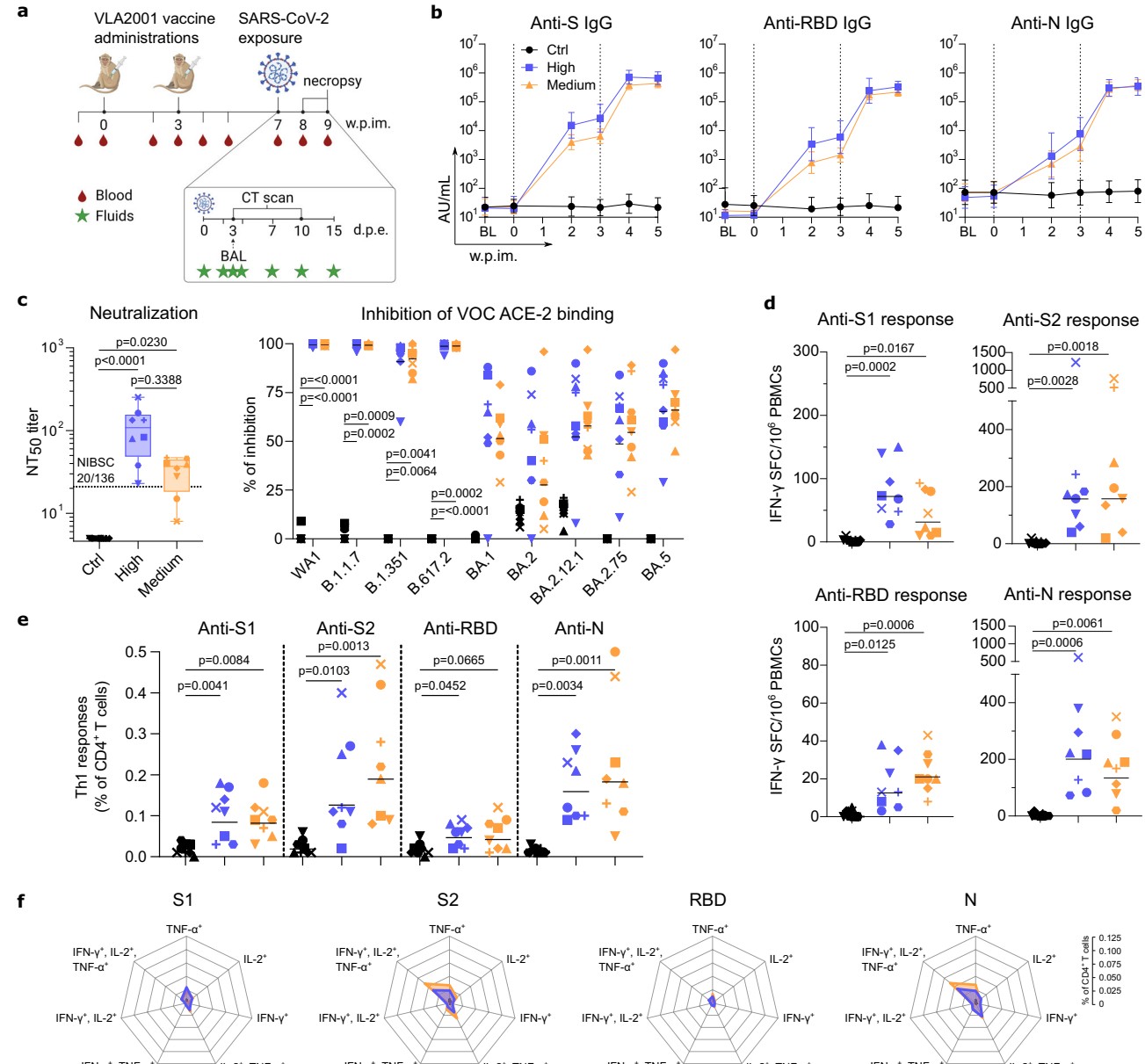

**Fig. 2 | Immunogenicity of VLA2001 in cynomolgus macaques. a** Study design.
**b** Geometric mean titer (AU/mL; error bars represent the 95% confidence interval)
of IgG binding to SARS-CoV-2 S, RBD and N proteins measured in sera samples
($n = 8$ biologically independent animals per group). BL: baseline approximately 3
weeks before first immunization. Dotted vertical lines represent vaccinations.
**c** Individual neutralizing titers to SARS-CoV-2 ($NT_{50}$: 50% neutralizing titer; Box
and whiskers, min to max) (non-parametric Kruskal-Wallis and Dunn's multiple
comparisons tests; $n = 8$ biologically independent animals per group) in sera sam-
pled 5 weeks following first immunization. Lower limit of quantification was 5.
Dotted horizontal line indicates the titer of the WHO International Standard for
anti-SARS-CoV-2 immunoglobulin in human determined in the same assay. Graph
starts at 4.6; Individual values (% of inhibition) of antibodies blocking the binding of
SARS-CoV-2 S protein to ACE2 using WA1, B.1.1.7, B.1.351, B.617.2 + AY.4 alt seq
2, B.1.1.529 + BA1, BA.2, BA.2.12.1, BA.2.75 and BA.5 antigens in sera sampled at
week 7 post first immunization. Horizontal bars represent the geometric means
(non-parametric Kruskal-Wallis and Dunn's multiple comparisons tests; $n = 8$
biologically independent animals per group). **d** Individual number of IFN-γ-
secreting cells analyzed by ELISpot after ex vivo overnight stimulation with SARS-
CoV-2 S1, S2, RBD or N overlapping peptide pools and plotted as spot-forming cells

(SFC) per $1 \times 10^6$ PBMCs, at week 5 post first immunization. Horizontal bars
represent the geometric means (non-parametric Kruskal-Wallis and Dunn's mul-
tiple comparisons tests; $n = 8$ biologically independent animals per group).
**e** Frequency of antigen-specific Th1 CD4+ T cells (CD154+ and IFN-γ+ +/- IL-2+
+/- TNF-α+) in the total CD4+ T cell population at week 5 post first immunization.
PBMCs were stimulated overnight with the SARS-CoV-2 S1, S2, RBD or N over-
lapping peptide pools. Horizontal bars represent the geometric means (non-para-
metric Kruskal-Wallis and Dunn's multiple comparisons tests; $n = 8$ biologically
independent animals per group). **f** Polyfunctionality of antigen-specific CD4+ T cells
(CD154+ or TNF-α+ or IL-2+ or IFN-γ+ and IL-2+/TNF-α+ or IFN-γ+/TNF-α+ or
IFN-γ+/IL-2+ and IFN-γ+/IL-2+/TNF-α+) in the total CD4+ T cell population at
week 5 post first immunization. PBMCs were stimulated overnight with SARS-CoV-
2 S1, S2, RBD or N overlapping peptide pools. Each intersection on the radar charts is
separated by 0.025. In (**b**), Control group is represented by black circle, High dose
group is represented by blue square and Medium dose group is represented by
orange triangle. In panels c to e, each animal is identified by distinct symbols
(Supplementary Table 1 for matching symbols). In (**c**–**f**), Control group is in black,
High dose group is in blue and Medium dose group is in orange.

different VOCs as evidenced by their capacity to inhibit the binding of the S protein (Fig. 2c) of the B.1.1.7 (Alpha), B1.351 (Beta) and B.617.2 (Delta) VOCs to human ACE2 receptor. Whereas cross-neutralization of Omicron variants appears reduced.

In addition to eliciting significant levels of specific antibodies, the VLA2001 vaccine induced detectable T cell specific responses. This was measured with an ex vivo IFN-γ ELISpot assay against the S protein S1 subunit, the S2 subunit, the RBD and the N antigens on PBMCs collected two weeks post boost and stimulated overnight with corresponding peptide pools (Fig. 2d). Further analysis of PBMCs by intracellular cytokine staining demonstrated that the response was predominantly CD4+-mediated Th1-biased and that the T cells specific for S2 and N exhibited a high degree of

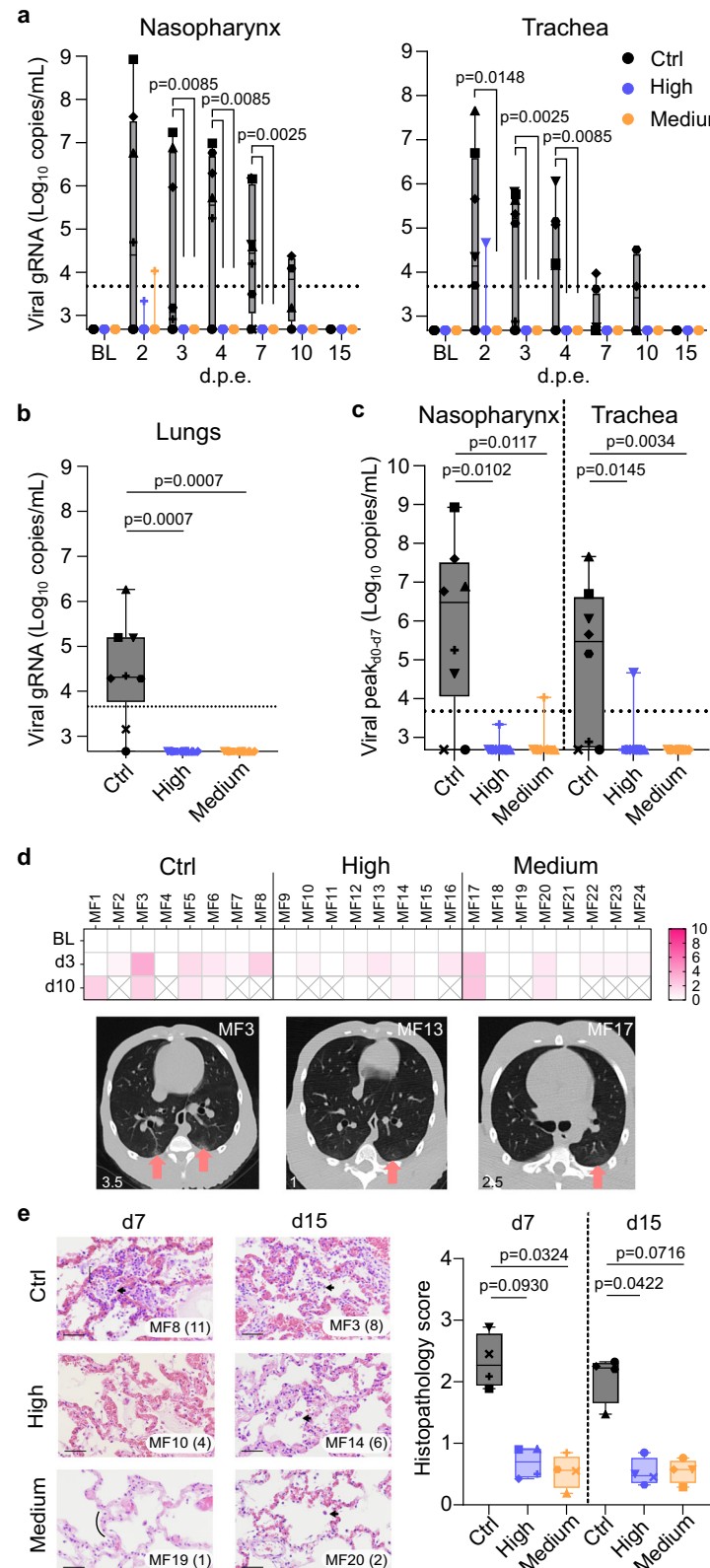

**Fig. 3 | Efficacy of VLA2001 vaccine in cynomolgus macaques. a** Individual genomic viral RNA (gRNA) quantification in nasopharyngeal and tracheal fluids collected with swabs (Box and whiskers, min to max) (non-parametric Kruskal-Wallis and Dunn's multiple comparisons tests, same *p* value for both analysis per time point when two lines and one *p* value appear; *n* = 8 biologically independent animals per group). Dotted horizontal line represents the lower limit of quantification (LLOQ) at 3.67 log$_{10}$ copies/mL. Graphs start at the lower limit of detection (LLOD) at 2.67 log$_{10}$ copies/mL. Values below the LLOD were plotted at 2.67 log$_{10}$ copies/mL. Data at BL on graphs are from specimen sampled at week 5 post first immunization. **b** Individual gRNA quantification in fluids from bronchoalveolar lavages (BAL) performed at day 3 post SARS-CoV-2 exposure (Box and whiskers, min to max) (non-parametric Kruskal-Wallis and Dunn's multiple comparisons tests; *n* = 8 biologically independent animals per group). Dotted horizontal line represents the LLOQ at 3.67 log$_{10}$ copies/mL. Graph starts at the LLOD at 2.67 log$_{10}$ copies/mL. Values below the LLOD were plotted at 2.67 log$_{10}$ copies/mL. **c** Peak of gRNA quantification in nasopharyngeal and tracheal fluids collected with swabs (Box and whiskers, min to max) (non-parametric Kruskal-Wallis and Dunn's multiple comparisons tests; *n* = 8 biologically independent animals per group). Dotted horizontal line represents the lower limit of quantification (LLOQ) at 3.67 log$_{10}$ copies/mL. Graphs start at the lower limit of detection (LLOD) at 2.67 log$_{10}$ copies/mL. Values below the LLOD were plotted at 2.67 log$_{10}$ copies/mL. **d** Heatmap of the CT score of pulmonary lesions detected by CT imaging performed at baseline (BL, week 5 post first immunization), day 3 and day 10 following SARS-CoV-2 exposure; Representative transversal slices of chest CT scans performed at day 3 post SARS-CoV-2 exposure showing typical lung lesions (arrows). **e** Histopathology slices and score of lungs at day 7 or day 15 post SARS-CoV-2 exposure. On the images, the associated score is indicated in parentheses. Arrows represent alveolar macrophages aggregates. The curly bracket represents interstitial lymphocytic infiltration. The curve represents a normal alveolus. The scale bar on each image represents 100 μm. Images were taken at ×40 magnification; On the graph, the mean of the 80 spots per animal is represented for every groups. For each animal, 20 spots were assessed on left and right caudal and cranial lobes. Each spot is the accumulation of the score associated to the 5 parameters described in Table S2 (Box and whiskers, min to max) (non-parametric Kruskal-Wallis and Dunn's multiple comparisons tests; *n* = 4 biologically independent animals per group). In (**a–c, e**), each animal is identified by distinct symbols (Supplementary Table 1 for matching symbols). In each of these panels, Control group is in black, High dose group is in blue and Medium dose group is in orange.

polyfunctionality (Fig. 2e, f). No CD8$^+$ T cell response was observed following immunizations.

### The VLA2001 vaccine prevented infection in cynomolgus macaques exposed with SARS-CoV-2

To assess VLA2001 vaccine efficacy, four weeks following the boost, all animals were exposed to a high dose ($1 \times 10^5$ pfu) of SARS-CoV-2 WA1 strain administered via a combined intra-nasal and intra-tracheal route using a previously reported challenge procedure[29–31,43]. Compared to the strain used to produce the vaccine, the challenge strain has a single amino acid difference in the S protein (V367F Mutation).

All eight control animals became infected as evidenced by the detection of genomic viral RNA (gRNA) or subgenomic viral RNA (sgRNA) in samples collected from nasopharyngeal and tracheal compartments and from broncho-alveolar lavages (BAL; Fig. 3a–c and Supplementary Fig. 3). In contrast, only one animal from Groups High and Medium had detectable gRNA in the upper respiratory tract at a single time point, very early (2 days) post exposure and with levels below or close to the limit of quantification (Fig. 3a). None of the vaccinated animals showed detectable sgRNA in upper or lower respiratory tract (Supplementary Fig. 3). Both vaccinated groups exhibited similar levels of protection against gRNA measured in BAL three days post exposure compared to control animals (Fig. 3b) and when comparing the peak of gRNA in the upper respiratory tract (Fig. 3c). Control animals displayed

an increase of anti-S and anti-N IgG within two weeks following exposure whereas both vaccinated groups still presented saturated IgG titers (Supplementary Fig. 1). The CD4$^+$ Th1-biased T cells from vaccinated animals remain polyfunctional after SARS-CoV-2 challenge whereas control animals exibited predominantly S- and N-specific IFN-γ-producing CD4$^+$ T cells (Fig. 4a). Correlation analysis between multiple assessed parameters demonstrated that the induction of B and T cell responses was negatively correlated to the detection of genomic viral RNA in lower respiratory tract (Figs. 4b, c). Following viral exposure, typical minor pulmonary lesions were observed by CT-scans in all groups with no statistical difference (Fig. 3d and Supplementary Fig. 4). The vaccine also protected from mild inflammatory responses observed in lung tissue of control NHPs euthanized at days 7 (four animals per group) or 15 (four animals per group) post exposure (Fig. 3e). The lesions observed in control group were typical of SARS-CoV-2 infection in NHPs and characterized with alveolar septal infiltration with immune and inflammatory cells such as lymphocytes, neutrophils and macrophages. Lesions in controls persisted 15 days post exposure. The Groups High and Medium showed a significant reduction ($p = 0.0930$ and $p = 0.0324$ for Groups High and Medium respectively at day 7 and $p = 0.0422$ and $p = 0.0716$ for Groups High and Medium respectively at day 15, non-parametric Kruskal-Wallis and Dunn's multiple comparisons tests; $n = 4$ per group) of lung lesions scoring (Fig. 3e) confirming protection against disease.

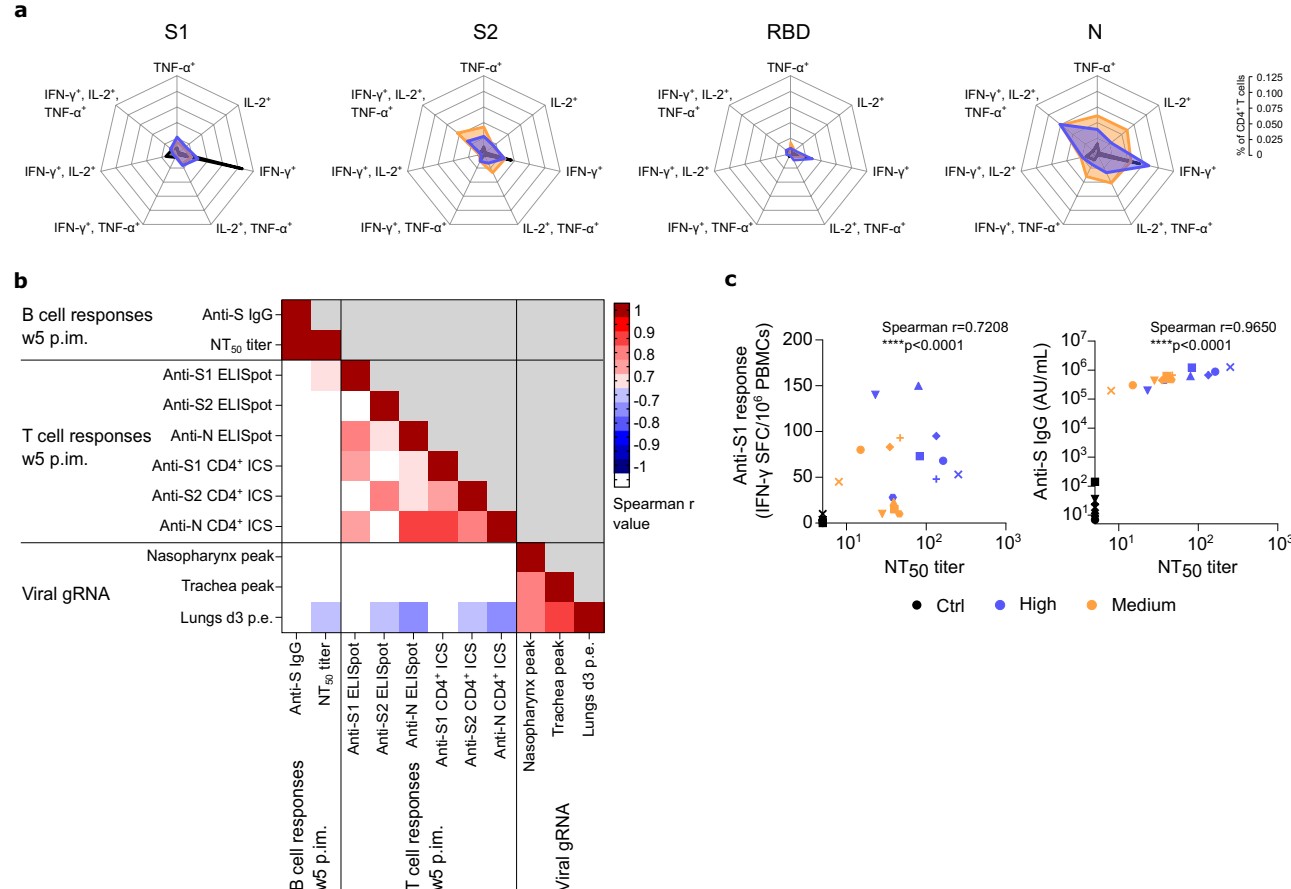

**Fig. 4 | T cell polyfunctionality and correlation analysis. a** Polyfunctionality of antigen-specific CD4$^+$ T cells (CD154$^+$ or TNF-α$^+$ or IL-2$^+$ or IFN-γ$^+$ and IL-2$^+$/TNF-α$^+$ or IFN-γ$^+$/TNF-α$^+$ or IFN-γ$^+$/IL-2$^+$ and IFN-γ$^+$/IL-2$^+$/TNF-α$^+$) in the total CD4$^+$ T cell population at day 7 post exposure. PBMCs were stimulated overnight with SARS-CoV-2 S1, S2, RBD or N overlapping peptide pools. Each intersection on the radar charts is separated by 0.025. **b** Correlation matrix between humoral, cellular and virological parameters. Spearman r values from -0.7 to -1 and 0.7 to 1 are

visible. **c** Correlation between ELISpot IFN-γ-secreting cells upon ex vivo overnight stimulation with SARS-CoV-2 S1 overlapping peptide pools and neutralizing antibody titers to SARS-CoV-2 at week 5 p.im. (w5); Correlation between IgG binding to SARS-CoV-2 S protein and neutralizing antibody titers to SARS-CoV-2 at w5 p.im. In panels a and c, Control group is in black, High dose group is in blue and Medium dose group is in orange. In (**c**), each animal is identified by distinct symbols (Supplementary Table 1 for matching symbols).

## Discussion

VLA2001 was the first COVID-19 vaccine to receive a standard marketing authorization in Europe and an Emergency Use Authorization (EUA) from the Kingdom of Bahrain and United Arab Emirates, for use as primary vaccination in people from 18 to 50 years of age. The preclinical studies described here extend the results of the phase III trial reported[44] which showed that VLA2001 was well tolerated, demonstrating a statistically significant better tolerability profile and induced superior neutralizing antibody titer levels when comparing VLA2001 to the active comparator vaccine, AstraZeneca's AZD1222. Broad antigen-specific IFN-γ-producing T cells were induced, and a neutralizing response was observed in over 95% of vaccinated volunteers. Similar responses observed in macaques were associated with almost complete protection from virus replication in the upper and lower respiratory tracts especially against first VOCs suggesting the vaccine would significantly control secondary transmission of the SARS-CoV-2. VLA2001 could therefore contribute to the control of virus circulation in vaccinated populations.

Alternative vaccine strategies in NHPs have already demonstrated comparable humoral and cellular responses as well as protection in upper and lower respiratory compartments[27–29,31,38–40,43]. Importantly, data obtained in our study confirmed a CD4$^+$-mediated Th1-biased antigen-specific response not only against the S protein, but also to the more conserved N protein, potentially improving the capacity to protect against emerging variants that have escape mutations in the S protein to evade neutralization. Thus, VLA2001 may have decisive advantages over Spike only based vaccines in seasonal waves of COVID-19. In such a context, the multicomponent aspect of the vaccine may be of significant importance to boost natural B and T cell immunity against all major viral antigens induced by previous exposure of the population to SARS-CoV-2 variants. This multicomponent hybrid response may also contribute to increased breadth of the induced immunity. Clinical and preclinical safety and efficacy data generated with this vaccine argue for an extension of the use of VLA2001 to young and aging populations, particularly in the context of boosting vaccine and/or natural preexisting immunity.

## Data availability

Source data underlying all figures can be found in the file 'Supplementary Data 1' uploaded with the manuscript. Raw data are included with a summary and one sheet per type of raw data. Each sheet is labeled by the figure number and panel.

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

## Acknowledgements

We thank Alexandra Skritek and Ingmar Stoll for support with the mouse studies. We thank Benoit Delache, Quentin Sconosciuti, Victor Magneron, Pauline Le Calvez, Maxime Potier, Jean-Marie Robert, Nina Dhooge, Manon Rimlinger, Jiahao Qiu, Thierry Prot, Kévin Tirmarche, Marjorie Benfissa and Cristina Dodan for the NHP experiments; Laetitia Bossevot, Laurine Moenne-Loccoz, Suzie Hecquet and Loïc Pintore for the RT-qPCR, ELISpot and MSD serology assays, and for the preparation of reagents; Mario Gomez-Pacheco and Jérôme van Wassenhove for the flow cytometry assays, Blanche Fert and Nidhal Kahlaoui for the in vivo imaging; Sophie Luccantoni and Camille Ludot for histology analysis. We thank Julia Pilot, Karen Storck, Céline Aubenque and Mickael Jegat for the MSD serology assays and neutralization assays; Elodie Guyon, Julien Dinh, Tracy Abauzit and Alexis Gomes for the NHP sample processing; Sylvie Legendre for the transports organization; Frédéric Ducancel, Alicia Pouget and Yann Gorin for the logistics and safety management; Pauline Maisonnasse for ethics management; Isabelle Mangeot and Camille Morice for help with resources management and Brice Targat and Karl-Stefan Baczowski who contributed to data management. We thank Sylvie Van der Werf, Sylvie Bellil and Vincent Enouf for contribution to viral stock challenge production and Antoine Nougairede for sgRNA assays calibration. Study design in Fig. 2a was performed on BioRender.com. The Infectious Diseases Models for Innovative Therapies (IDMIT) research infrastructure is supported by the "Programme d'investissements d'avenir", managed by the ANR under reference ANR-11-INBS-0008. The NHP model of SARS-CoV-2 infection have been developed thanks to the support from REACTing (ANRS-MIE) and the "Fondation pour la Recherche Médicale" (FRM; AM-CoV-Path). Valneva received funding from the Secretary of State for Business, Energy and Industrial Strategy, UK.

## Author contributions

R.S., J.H.W.: Contributed to the vaccine production, characterization, analyses, and formulation. U.L., S.S., U.B., A.M.: Contributed to mouse study design, realization, and analyses. U.L., A.M.: Contributed to manuscript writing. M.G., R.L.G.: Contributed to NHP study design, realization and analysis, and contributed to manuscript writing; R.M.: Contributed to NHP experiments and data analysis. F.R., S.L., J.L.: Contributed to NHP experiment, data generation and analysis. N.D.B., J.M., M.G.L., A.S.G., W.G., C.C., Q.P., T.N., M.C., V.C.: Contributed to NHP data generation and analysis. K.M., G.F., N.L.: Designed and analyzed the anti-SARS-CoV-2 binding and neutralizing IgG response bioanalysis in NHP sera. All authors reviewed the manuscript.

## Competing interests

The authors declare the following competing interests: U.L., R.S., J.H.W., S.S., U.B., A.M. are employees of Valneva Austria GmbH. K.M., G.F., N.L. are employees of Oncodesign Services. The other authors declare no competing interest.

## Additional information

**Peer review information** : *Communications Medicine* thanks Ed Remarque and the other, anonymous, reviewer(s) for their contribution to the peer review of this work. A peer review file is available.

[1]Center for Immunology of Viral, Auto-immune, Hematological and Bacterial diseases (IMVA-HB/IDMIT), Université Paris-Saclay, Inserm, CEA, Fontenay-aux-Roses, France. [2]VALNEVA Austria GmbH, Vienna, Austria. [3]ONCODESIGN SERVICES, François Hyafil Research Center, Villebon-sur-Yvette, France. [4]These authors contributed equally: Andreas Meinke, Roger Le Grand. ✉e-mail: Andreas.MEINKE@valneva.com; roger.le-grand@cea.fr

