## [Peer Review File · Communications Medicine]

Reviewers' comments:

Reviewer #1 (Remarks to the Author):

Manuscript Galhaut et al. arises an important issue of inactivated vaccine use in post-pandemic era as they are less reactogenic, have less limitations for use, expected long-term consequences, low production cost and their antigen can be easily changed to a currently circulating virus variant to be used as a heterologous booster after primary vaccination course with any vaccine available or prepared as a combined vaccine with several antigens (of the same virus or other). For these reasons, all the data on spectrum of immunity and protection of inactivated vaccines should be presented. Even, the pre-clinical data is valuable if the vaccine has passed the clinical trials and being used for vaccination.

Overall, the data presented is clear, experiments are done nicely, data presented in full.

However, during the work the VLA2001 vaccine induced immunity was tested against rather old SARS-CoV-2 variants long substituted from the circulation by Omicron derivatives (currently XBB lineages): antibody levels were determined against WA.1, B.1.17, B.1.351 and P1 pseudoviruses, anti-S and anti-RBD IgGs were determined against old antigens as well, etc.

The same goes with protective experiment in macaques - WA.1 strain was used as a challenge virus, which is antigenically similar to the vaccine antigen. So, the experiment was performed with homologous challenge.

Therefore, pseudotype virus neutralization test with mice/macaque sera using current SARS-CoV-2 variants would add more value to the study.

Minor comments:

- L64-66 - please, add the Pango lineage for the vaccine strain
- L70 - what does antigen unit correspond to? in mcg or reference to the experimental procedure and calculation
- Figure 1 - graphs g and h - please add titles (anti-S and NTs) as on other graphs

Reviewer #2 (Remarks to the Author):

High efficacy of VLA2001 vaccine against SARS-CoV-2 infection in non-human primates.

Mathilde Galhaut. et al.

General:

The paper describes an inactivated adjuvanted whole virus SARS-CoV-2 vaccine. Humoral and cellular immunogenicity are demonstrated in mice. Humoral and cellular immunogenicity as well as protective capacity following homologous challenge are demonstrated in cynomolgus macaques. There are a number of papers published on the safety and efficacy SARS-CoV-2 vaccines in macaques, the current paper is the first reporting on an adjuvanted inactivated whole virus vaccine, the results should however be discussed in relation to other publications in macaques.

Specific comments:

Lines 41-42. It should be noted that the frequency of the adverse events mentioned was of low frequency.

Line 43: "Second, several of the available vaccines utilize one or two SARS-CoV-2 Spike (S) proteins, to elicit protective immunity."

It is not clear to me what is meant here, do the authors refer to the antigenic variants in the bivalent mRNA vaccines? Please clarify.

Line 44: "Although the efficacy of these vaccines was high against the ancestral virus and remained

high against several variants of concern (VOC)[11], it dropped precipitously with the emergence of the omicron VOC[12-17], which has a S protein sequence that is much more divergent from the ancestral SARS-CoV-2 than previous VOCs.”

The efficacy of qRT-PCR-confirmed infection is indeed much lower for the Omicrons, yet efficacy against severe disease and hospitalisation is still quite high. This should be mentioned.

Line 50: “Inactivated vaccines have proven their efficacy against many pathogens and have good safety records in all populations, including children, pregnant women, and the elderly.”

This is no guarantee whatsoever that the vaccine under investigation will be safe. Perhaps better to remove this statement.

Lines 64-91: It would be worthwhile to have information on the vaccine dose other than arbitrary units. Can the authors indicate the amount of virus protein in the vaccine. For Covaxin the amount of virus protein is known to be 6 µg.

Lines 98-99: Why was PBS chosen as control vaccine and not AIOH-CpG with an irrelevant antigen or plain adjuvant?

Lines 114-116: The Beta and Gamma lineage Spike proteins are not very different from the vaccine antigens, whereas the Omicron Spike are very much different. It would be interesting to also have neutralisation titres for Omicron variants.

Lines 120-123: Did the authors observe CD8+ responses? If not please mention this.

Lines 125-127: Can the authors provide information on the antigenic differences between the spike proteins of the vaccine and challenge antigen? How homologous are those?

Lines 134-136: A comparison of AUC was performed for gRNA. I’m not sure whether this is methodically correct as the sampling was on days 2, 3, 4 & 7 and the non-sampled days are interpolated, thereby generating data that were not observed. It is better to just sum the observed values and compare those.

Lines 158-159: This only holds for the original isolate and not for later variants in particular the Omicrons.

Lines 285-287: Please provide more information about the cynomolgus macaques: What was their origin (Mauritius?), what were the body weights, what were the inclusion / exclusion criteria, what was their virus status (Herpes B, SRV, STLV etc.), how many animals were screened etc. (see Arrive guidelines). Can the authors be more specific on the randomisation procedure?

Lines: 301-306: Can the authors provide information on the vaccine batch, i.e. was the GMP vaccine used in these experiments and if not what was the quality of the vaccine batch (e.g. pre-GMP run).

Lines 342-347: It would be helpful to have a table with the characteristics of the monkeys in the treatment groups see comments for lines 285-287.

Lines 355-357: Was the selection of the animals to be euthanised on day 7 randomised and was this done before the challenge?

Lines 430-432: For the BAL procedure the amount of liquid used may have diluted the amount of virus. How reproducible is the BAL procedure and how much fluid was retrieved?

Lines 459-463: Was the pathologist blinded for treatment?

Lines 465-467: As far as I know the Tukey test is named Tukey’s Honest Significant Difference. I’m not sure whether a two-way ANOVA was used as only treatment is reported as explanatory variable. When reporting p-values please indicate which test was employed, i.e. ANOVA or Kruskal-Wallis.

Figure 2:

Panel A. please also include the Bal at day 3 post challenge.

Concerning colour and symbol shapes: It would be helpful if individual animals within a treatment group could be identified across panels by using shapes and or colours such that the reader can see whether a high S1 response is also present in the same animal for RBD and S2 etc.

Figure 3B and E. What do the error bars represent and what kind of statistical test was used? If the

statistical test was non-parametric it is better to provide a box-whiskers plot.

Extended data:

Figure S1. The control animals show increases in Anti S and RBD at week 9, whereas the vaccinated animals do not. Can the authors provide an explanation for this observation?

Figure S3. What do the error bars represent and what kind of statistical test was used? If the statistical test was non-parametric it is better to provide a box-whiskers plot.

Response to Referee #1

Manuscript Galhaut et al. arises an important issue of inactivated vaccine use in post-pandemic era as they are less reactogenic, have less limitations for use, expected long-term consequences, low production cost and their antigen can be easily changed to a currently circulating virus variant to be used as a heterologous booster after primary vaccination course with any vaccine available or prepared as a combined vaccine with several antigens (of the same virus or other). For these reasons, all the data on spectrum of immunity and protection of inactivated vaccines should be presented. Even, the pre-clinical data is valuable if the vaccine has passed the clinical trials and being used for vaccination. Overall, the data presented is clear, experiments are done nicely, data presented in full. However, during the work the VLA2001 vaccine induced immunity was tested against rather old SARS-CoV-2 variants long substituted from the circulation by Omicron derivatives (currently XBB lineages): antibody levels were determined against WA.1, B.1.17, B.1.351 and P1 pseudoviruses, anti-S and anti-RBD IgGs were determined against old antigens as well, etc. The same goes with protective experiment in macaques - WA.1 strain was used as a challenge virus, which is antigenically similar to the vaccine antigen. So, the experiment was performed with homologous challenge. Therefore, pseudotype virus neutralization test with mice/macaque sera using current SARS-CoV-2 variants would add more value to the study.

We thank **Referee #1** for their positive and insightful comments. Concerning the antibody response against Omicron variants, we have performed an ACE-2 competition assay with a commercial kit including BA.1, BA.2 and BA.5 antigens. The results were added and provide a more complete analysis.

Minor comments:

L64-66 - please, add the Pango lineage for the vaccine strain.

- The vaccine strain is Pango lineage B. It has been added lines 66 (in “Main text” section) and 314 (in “Methods” section).

L70 - what does antigen unit correspond to? in mkg or reference to the experimental procedure and calculation.

- The dosage ranging from 0.3 to 35 antigen units corresponds to 0.02 to 2.1 µg total viral protein. It has been added lines 70-71 (in “Main text” section).

Figure 1 - graphs g and h - please add titles (anti-S and NTs) as on other graphs.

- The titles of the Figures 1g and 1h were added.

High efficacy of VLA2001 vaccine against SARS-CoV-2 infection in non-human primates. Mathilde Galhaut. et al.

General:

The paper describes an inactivated adjuvanted whole virus SARS-CoV-2 vaccine. Humoral and cellular immunogenicity are demonstrated in mice. Humoral and cellular immunogenicity as well as protective capacity following homologous challenge are demonstrated in cynomolgus macaques. There are a number of papers published on the safety and efficacy SARS-CoV-2 vaccines in macaques, the current paper is the first reporting on an adjuvanted inactivated whole virus vaccine, the results should however be discussed in relation to other publications in macaques.

We thank **Referee #2** for their thorough review that will improve our manuscript. We answered to all comments and the results have been highlighted in relation to other publications in macaques as suggested (see lines 168-170 in the main text).

Specific comments:

Lines 41-42. It should be noted that the frequency of the adverse events mentioned was of low frequency.

→ Sure, these adverse events have been observed in rare cases. It has been mentioned line 41 (in “Main text” section).

Line 43: “Second, several of the available vaccines utilize one or two SARS-CoV-2 Spike (S) proteins, to elicit protective immunity.” It is not clear to me what is meant here, do the authors refer to the antigenic variants in the bivalent mRNA vaccines? Please clarify.

→ Yes, this comment refers to the bivalent mRNA vaccines. It has been clarified line 44 (in “Main text” section).

Line 44: “Although the efficacy of these vaccines was high against the ancestral virus and remained high against several variants of concern (VOC)[11], it dropped precipitously with the emergence of the omicron VOC[12-17], which has a S protein sequence that is much more divergent from the ancestral SARS-CoV-2 than previous VOCs.” The efficacy of qRT-PCR-confirmed infection is indeed much lower for the Omicrons, yet efficacy against severe disease and hospitalisation is still quite high. This should be mentioned.

→ Indeed, these vaccines are effective against severe disease and hospitalization. It has been specified line 48 (in “Main text” section).

Line 50: “Inactivated vaccines have proven their efficacy against many pathogens and have good safety records in all populations, including children, pregnant women, and the elderly.” This is no guarantee whatsoever that the vaccine under investigation will be safe. Perhaps better to remove this statement.

→ Your comment is pertinent, it has been removed.

Lines 64-91: It would be worthwhile to have information on the vaccine dose other than arbitrary units. Can the authors indicate the amount of virus protein in the vaccine. For Covaxin the amount of virus protein is known to be 6 µg.

- The dosage ranging from 0.3 to 35 antigen units corresponds to 0.02 to 2.1 µg total viral protein. It has been added lines 70-71 (in “Main text” section).

Lines 98-99: Why was PBS chosen as control vaccine and not AIOH-CpG with an irrelevant antigen or plain adjuvant?

- We choose the PBS as control vaccine in order to match with other NHP COVID studies in the lab and compare with historical controls.

Lines 114-116: The Beta and Gamma lineage Spike proteins are not very different from the vaccine antigens, whereas the Omicron Spike are very much different. It would be interesting to also have neutralisation titres for Omicron variants.

- We have performed an ACE-2 competition assay against Omicron variants and the results were included in the Figure 2c (right panel). We observed a lower inhibition than Alpha, Beta and Delta variants. It also has been discussed lines 119-120 in the main text. The protocol has been corrected from line 403 to 419 (in “Methods” section). The legend of the Figure 2c has been modified from line 526 to 528 (in “Legends” section).

Lines 120-123: Did the authors observe CD8+ responses? If not please mention this.

- We did not observed any CD8⁺ T cell response following immunizations. It has been mentioned line 127 (in “Main text” section).

Lines 125-127: Can the authors provide information on the antigenic differences between the spike proteins of the vaccine and challenge antigen? How homologous are those?

- The vaccine derived from the strain hCoV-19/Italy/INMI1-isl/2020, whereas the challenge was performed with the strain BetaCoV/France/IDF/0372/2020. The two strains are highly homologous with only a single amino acid difference in the spike protein (V367F in the Spike protein of the French strain compared to Italian strain). It has been specified lines 131-133 (in “Main text” section).

Lines 134-136: A comparison of AUC was performed for gRNA. I'm not sure whether this is methodically correct as the sampling was on days 2, 3, 4 & 7 and the non-sampled days are interpolated, thereby generating data that were not observed. It is better to just sum the observed values and compare those.

- We have modified Figures 3a, 3c and S3a. Instead of showing viral gRNA with curves in Figure 3a, the data were represented by histograms displaying only the time-points of the study. In figures 3c and S3a, we have replaced the comparison of AUC by the comparison of peak of infection. The legends of these figures and the main text (line 141) were also modified. The legend have been modified for Figure 3c in line 551 (in “Legends” section). In the Supplementary Information file, the legends have been modified for Figure S3a in line 12.

Lines 158-159: This only holds for the original isolate and not for later variants in particular the Omicrons.

- Sure, these responses have been especially observed against first VOCs. It has been specified line 165 (in “Main text” section).

Lines 285-287: Please provide more information about the cynomolgus macaques: What was their origin (Mauritius?), what were the body weights, what were the inclusion / exclusion criteria, what was their virus status (Herpes B, SRV, STLV etc.), how many animals were screened etc. (see Arrive guidelines). Can the authors be more specific on the randomisation procedure?

- The cynomolgus macaques are of Mauritius origin and it has been mentioned line 297 (in “Methods” section). The body weight and the age at the first immunization were added in the Table S1. Their virus status and the information about inclusion criteria and screening were described from line 306 to 310 (in “Methods” section).
Animals were randomly distributed into each experimental group using ALEA function of Excel software. It has been specified lines 358-359 (in “Methods” section).

Lines: 301-306: Can the authors provide information on the vaccine batch, i.e. was the GMP vaccine used in these experiments and if not what was the quality of the vaccine batch (e.g. pre-GMP run).

- The vaccine was GMP manufactured indeed. It has been cited line 316 (in “Methods” section).

Lines 342-347: It would be helpful to have a table with the characteristics of the monkeys in the treatment groups see comments for lines 285-287.

- The Table S1 was added and described the characteristics of the animals included into the study. It has been cited line 359 (in “Methods” section).

Lines 355-357: Was the selection of the animals to be euthanised on day 7 randomised and was this done before the challenge?

- The selection of the animals to be euthanized on days 7 or 15 was performed before the first immunization during the randomized distribution into each experimental group. It has been mentioned lines 374-375 (in “Methods” section).

Lines 430-432: For the BAL procedure the amount of liquid used may have diluted the amount of virus. How reproducible is the BAL procedure and how much fluid was retrieved?

- We performed BAL in every studies (more than 30) in which animals were exposed to SARS-COV-2 with the same protocol and we collected an average of 30 mL of fluids for all the studies. We perform the BAL with a fixed volume of 50 mL of 0.9 % NaCl and we collect a mean of 30 mL of fluids. During the study, the collected volume varied from 25 to 35 mL. It has been modified in line 451 (in “Methods” section).

Lines 459-463: Was the pathologist blinded for treatment?

- Yes, the pathologist was blinded for treatment and it has been added lines 479-480 (in “Methods” section).

Lines 465-467: As far as I know the Tukey test is named Tukey’s Honest Significant Difference. I’m not sure whether a two-way ANOVA was used as only treatment is reported as explanatory variable. When reporting *p*-values please indicate which test was employed, i.e. ANOVA or Kruskal-Wallis.

- The Tukey test was a mistake. It has been removed from lines 485-486. The use of the Kruskal-Wallis and Dunn’s multiple comparisons tests has been mentioned lines 102-103 and 106-107.

Figure 2:

Panel A. please also include the Bal at day 3 post challenge. Concerning colour and symbol shapes: It would be helpful if individual animals within a treatment group could be identified across panels by using shapes and or colours such that the reader can see whether a high S1 response is also present in the same animal for RBD and S2 etc.

- The BAL at day 3 post challenge has been added in the Figure 2a. Each animal has been identified by a unique symbol and color in all graphs showing individual values. The corresponding symbol for each animal has been also indicated in the Table S1.

Figure 3B and E. What do the error bars represent and what kind of statistical test was used? If the statistical test was non-parametric it is better to provide a box-whiskers plot.

- All histograms (Figures 2c, 3a, 3b, 3c, 3e, S3a and S3b) have been replaced by box-whiskers plot. The legend have been modified for Figure 2c in line 522, Figure 3a in lines 541-542, Figure 3b in line 548, Figure 3c in line 552 and Figure 3e in lines 564-565 (in “Legends” section). In the Supplementary Information file, the legends have been modified for Figure S3a in line 13 and Figure S3b in line 18.

Extended data:

Figure S1. The control animals show increases in Anti S and RBD at week 9, whereas the vaccinated animals do not. Can the authors provide an explanation for this observation?

- The vaccinated animals reached saturated IgG titers before exposure whereas the control animals increased their IgG titers only after virus exposure. This observation has been explained from lines 142-143 (in “Main text” section).

Figure S3. What do the error bars represent and what kind of statistical test was used? If the statistical test was non-parametric it is better to provide a box-whiskers plot.

- All histograms (Figures 2c, 3a, 3b, 3c, 3e, S3a and S3b) have been replaced by box-whiskers plot. The legend have been modified for Figure 2c in line 522, Figure 3a in lines 541-542, Figure 3b in line 548, Figure 3c in line 552 and Figure 3e in lines 564-565 (in “Legends” section). In the Supplementary Information file, the legends have been modified for Figure S3a in line 13 and Figure S3b in line 18.

Reviewers' comments:

Reviewer #1 (Remarks to the Author):

Authors addressed all the comments and added interesting data. The manuscript is ready for publication.

Reviewer #2 (Remarks to the Author):

High efficacy of VLA2001 1 vaccine against SARS-CoV-2 infection in non-human primates

Galhaut et al. Revised version.

The manuscript has been improved and most comments were addressed appropriately. The figures are also well adapted as per my requests.

I have a few remaining minor comments:

Summary:

Line 28: "polyfunctional T cell responses, predominantly Th1-biased" better to say: "polyfunctional CD4+ T cell responses, predominantly Th1-biased"

Line 31: "to a high dose of SARS-CoV-2" better to say: "to a high dose of homologous SARS-CoV-2"

Main text:

Lines 98-99: "male cynomolgus macaques injected either with a high dose (35 AU, Group High) or medium dose (7 AU, Group Medium) by intramuscular (i.m.) route on days 0 and 21 (Fig. 2a)."

Better to say: "male cynomolgus macaques injected either with a high dose (35 AU corresponding to 2.1 µg of total virus protein, Group High) or medium dose (7 AU corresponding to 0.3 µg of total virus protein, Group Medium) by intramuscular (i.m.) route on days 0 and 21 (Fig. 2a)."

Line 358: There is no Excel function called ALEA to my knowledge. Excel has the RAND function to generate uniform random numbers. Please correct.

Response to Referee #1

Authors addressed all the comments and added interesting data. The manuscript is ready for publication.

We thank **Referee #1** for validating our corrections.

Referee #2

The manuscript has been improved and most comments were addressed appropriately. The figures are also well adapted as per my requests.

We thank **Referee #2** for validating our previous corrections and for his minor comments which helped improve our manuscript.

I have a few remaining minor comments:

Summary:

Line 28: “polyfunctional T cell responses, predominantly Th1-biased” better to say: “polyfunctional CD4+ T cell responses, predominantly Th1-biased”

→ It has been corrected line 28.

Line 31: “to a high dose of SARS-CoV-2” better to say: “to a high dose of homologous SARS-CoV-2”

→ It has been added line 31.

Main text:

Lines 98-99: “male cynomolgus macaques injected either with a high dose (35 AU, Group High) or medium dose (7 AU, Group Medium) by intramuscular (i.m.) route on days 0 and 21 (Fig. 2a).” Better to say: “male cynomolgus macaques injected either with a high dose (35 AU corresponding to 2.1 µg of total virus protein, Group High) or medium dose (7 AU corresponding to 0.3 µg of total virus protein, Group Medium) by intramuscular (i.m.) route on days 0 and 21 (Fig. 2a).”

→ The two specifications have been added to the two doses, lines 98-99 and 99 respectively.

Line 358: There is no Excel function called ALEA to my knowledge. Excel has the RAND function to generate uniform random numbers. Please correct.

→ It has been corrected line 359. ALEA is the French equivalent of the RAND function that we use in our Excel software.